# Weather Classification by Utilizing Synthetic Data

**DOI:** 10.3390/s22093193

**Published:** 2022-04-21

**Authors:** Saad Minhas, Zeba Khanam, Shoaib Ehsan, Klaus McDonald-Maier, Aura Hernández-Sabaté

**Affiliations:** 1School of Computer Science & Electrical Engineering, University of Essex, Wivenhoe Park, Colchester CO4 3SQ, UK; zeba.khanam@essex.ac.uk (Z.K.); sehsan@essex.ac.uk (S.E.); kdm@essex.ac.uk (K.M.-M.); 2Computer Vision Centre, Universitat Autònoma de Barcelona, Plaça Cívica, 08193 Bellaterra, Spain; aura@cvc.uab.cat; 3Departament de Ciències de la Computació, Universitat Autònoma de Barcelona, 08193 Bellaterra, Spain

**Keywords:** weather classification, synthetic data, dataset, autonomous car, computer vision, advanced driver assistance systems, deep learning, intelligent transportation systems

## Abstract

Weather prediction from real-world images can be termed a complex task when targeting classification using neural networks. Moreover, the number of images throughout the available datasets can contain a huge amount of variance when comparing locations with the weather those images are representing. In this article, the capabilities of a custom built driver simulator are explored specifically to simulate a wide range of weather conditions. Moreover, the performance of a new synthetic dataset generated by the above simulator is also assessed. The results indicate that the use of synthetic datasets in conjunction with real-world datasets can increase the training efficiency of the CNNs by as much as 74%. The article paves a way forward to tackle the persistent problem of bias in vision-based datasets.

## 1. Introduction

Simulation and virtual testing will play an increasingly relevant role, as they provide a very effective way to deal with the high number of scenarios Connected and Automated Driving (CAD) vehicles will encounter [1]. Road accidents related to adverse weather conditions play a huge part in disrupting the flow of traffic in a busy city environment [2,3,4]. The data available at present contains a large amount of variation. Figuring out a particular weather condition is a straight forward task for a normal human being but can be quite challenging for a computer vision system [5,6,7]. To overcome the challenges, neural networks, in recent decades, have revolutionized computer vision systems to detect the weather condition using images as an input. Indeed, Convolutional Neural Networks (CNN) have been deployed in various fields such as ship detection [8,9,10,11,12,13], object tracking in endoscopic vision [14,15], nuclear plant inspection [16,17,18], transport systems [19,20], and other complex engineering tasks [21,22]. Yet, there is still a lot of ground to cover. In the case of weather recognition on roads, the main challenges are: variability in elements such as camera placement and road layouts [23] and the machine learning methods such as CNN. Under such circumstances, there is a need to explore more methods of filling the gaps in between real world images; ideally, a set of images recorded in the same location but with different weather conditions would maximize the efficiency of a machine learning system. This is the main reason why the use of synthetic data can be more productive as compared to the real-world counterpart. In this paper, there are two main objectives: the first is to assess the modifications made to the driver simulator, which was previously used for driver vehicular interactions [24]. The second objective is to test the performance of the generated dataset with other comparable groundtruth datasets.

A custom-built virtual simulator that specializes in varying weather systems was previously implemented [25]. It utilizes Unity3d to simulate the weather with accurate lighting effects. The simulator environment was based on the real-world location of central Colchester, United Kingdom. It features a good mix of wide-open road and inner-city roads. An autonomous car was driven at different hours of the day in weather conditions ranging from sunny, cloudy, or rainy to foggy with different camera angles. The final dataset comprises 108,333 images, with approx. 35,000 images per class. The results show that state-of-the the art CNN architectures trained on the synthetic dataset were able to achieve an accuracy as high as 74% when tested on a real-world dataset [26].

The paper is organized in four sections. In Section 1, the main topic of the article is introduced in the light of recent advances made in this field, the simulator used, and the datasets used for weather classification training and testing. Section 2 presents a detailed description of the weather classification pipeline and the methodologies. Section 3 evaluates the experimental results and inferences that can be drawn. Finally, Section 4 concludes the research work presented in this article that can be used as a guide for future works.

### 1.1. Related Work

Most of the previous research includes the use of polarized and infrared cameras. The use of such cameras can provide some plausible data, but the installation costs can easily be substantial [27]. In order to overcome this issue, the use of RGB cameras is preferred because they are simpler and cost-effective, hence making it viable for mass production.

A study performed by Omer and Fu [28] used color cues to add illumination variance; however, their approach required the detection of white road lines to detect the road area, which can be quite challenging in severe winter weather conditions.

Most of the studies aimed toward driver assistance systems have been performed with regard to rainy weather classifications [29,30]. A study performed by Lu et al. [31] dealt with two class weather classification which included sunny and cloudy. In that study, the authors proposed a new data augmentation scheme to substantially enrich the training data, which was then used to train a latent SVM framework to make the solution insensitive to global intensity transfer. Another study [32] dealt with multi-class weather classification, which only used fixed camera points.

With regard to synthetic datasets, there is a lot of research being carried out to fill the gaps between real world data with synthetic data. There are some driver simulators that can fill in the void by generating synthetic datasets for weather classification. CARLA [33] is one such simulator that aides in autonomous research. It comprises a built in weather system that can be used to generate weather classification datasets. The Synthia dataset [34] is another example, which comprises 200,000 plus images with varying dynamic situations such as a clear sky, rain, and night. Hao et al. [35] developed a weather simulator that could replicate the weather at a given time in a virtual environment. However, it lacked visual fidelity for our experiments. The article also had specific requirements for the camera position’s location on the driver’s car.

Another plausible direction of the research into weather monitoring has been the use of microwave-based Synthetic Aperture Radar (SAR) imaging. Unlike optical sensors, this tool is unaffected by weather conditions. This is the main reason that they have been used for high speed ship detection [36,37]. The SAR images are used as input to a grid convolutional neural network (G-CNN) to detect ships and their speeds. Another prominent work has used a depthwise separable convolution neural network (DS-CNN) to detect high speed ships [38]. Another direction of research has focused on small sized ships [39] and the unperceived imbalance problem [40]. However, connection to a satellite is not always possible; thus, in this research, optical sensors were considered.

### 1.2. Contributions

Currently, autonomous driving cars deploy a series of sensors to classify the weather condition. However, this solution is not holistic and requires additional computational resources in terms of power and memory. More often, each weather type requires a specific sensor. For instance, rain is detected using a humidity sensor that plays no role in detecting sunny weather. Thus, using a camera to capture images and detecting weather using a single image is not only economical but also computationally less expensive. Finally, to our knowledge, most of these datasets are not aimed toward weather specifically.

For all these reasons the main technical contributions of this paper are the following:Holistic Weather Classification Solution. The solution proposed in this paper is a holistic solution in the sense that has been built from the ground up focusing on weather simulations;Simulator-based Training Dataset. Detection of weather using a single image requires a large images dataset with varying features for training CNNs, which turns out to be a bottleneck. In this work, a synthetic dataset for training CNNs is proposed;Real-Time Evaluation. The state-of-the-art CNNs are evaluated on real-time testing datasets to detect the accuracy of weather classification. Moreover, our classification approach differs from others in the sense that instead of processing only a portion of the image, it processes the entire image. This specific approach allows the system to record subtle changes in light intensity and color variations, which can be crucial in distinguishing between different weather conditions, such as cloudy and sunny.

### 1.3. The Simulator

The simulator that this study utilizes was previously used for the study of driver vehicular interaction. Ref [25] as shown in the Figure 1. Although it is capable of recording a driver’s behavior in detail, the setup was altered for this study. In particular, it is capable of recording multi-camera vehicle viewpoints. It was modified to reflect the real-world location of Colchester, United Kingdom. This includes a good variation of two-lane as well as single-lane roads. It also comprises a highly detailed interior as shown in Figure 2 as well as exterior, so the simulator capabilities can be extended even further if required for future studies. Moreover, a virtual camera was fitted to the top of the windshield just above the rear view mirror to better capture the environment ahead. Figure 3 shows the autonomous car that was used to record the necessary viewpoints for this study.

The virtual environment was roughly based on the small town of Colchester located in the Eastern United Kingdom. The initial goal was to construct a virtual motorway section based on the M25, but that was later amended for a more varied road environment. The total travel distance was over one mile, which consisted of a fair balance of double lane roads, roundabouts, and inner town single lane roads as shown in Figure 4.

The virtual world also contained a wide range of randomly generated traffic cars that added to the realism and complexity of its real-world counterpart. The simulator was designed in such a way that it provided a considerable amount of modification control for the researcher. The goal was to make it accessible for anyone to start generating varied weather conditions for autonomous research. In addition, it has the ability to adjust for a virtually unlimited number of weather variations and the amount of images that can be generated at any given time.

### 1.4. The Dataset

The generated dataset provides a plausible amount of varied weather conditions. The main classes include sunny, cloudy, foggy, and rainy. Each class then contains further subclasses involving the same class captured every hour from 9 AM to 4 PM. This methodology provided the most efficient learning material for CNNs and deep learning algorithms, as it provides the same location within varied lighting and weather conditions. For each recording session, the virtual car was allowed to run through the circuit, which resulted in the capture of approximately 2600 images. Each session was recorded on an hourly basis, i.e., for a clear day, weather for each session of driving was captured at 9:00, 10:00, 11:00, 12:00, 1:00, 2:00, 3:00, and 4:00. This provided a much needed variation in the overall shadow and lighting conditions for a varied dataset generation. Figure 5 shows the four main classes captured at various locations through different sessions. Table 1 shows the distribution of images per class. The resolution of each image was recorded at 1280 × 720; the channels used were red, green, and blue. Notice that the validation images for the foggy class only consist of five images; this is because a foggy image is by far the most specific in color tone and channel information. Moreover, the quantity of validation images was set by the creators of the Berkeley Deep Drive dataset.

Moreover, extensive care was taken to simulate secondary imperfections such as water droplets on the camera lens for distortion, traffic car signal bloom effects, and water shower behind traffic car wheels. Additional camera angles, such as left view, right view, and back view, were also captured to meet the challenge of the diverse task and absence of discriminatory features among various weather conditions.

Our synthetic dataset was evaluated on the Berkeley Deepdrive dataset [26], because it provides a considerable variation of varying weather conditions in a fairly balanced annotated pattern as shown in the Figure 6.

## 2. Weather Classification

To check to what extent our synthetic dataset was useful for weather classification, we applied a number of deep learning networks to test the dataset. One of the most famous deep learning architectures, Convolutional Neural Networks (CNN) have been able to perform various vision tasks with capabilities comparable to humans. However, CNN’s performance is highly dependent upon the large size of the training data. This problem intensifies for a weather classification task as the real-time weather variation data availability for self-driving cars is difficult [41] to attain. Based on this problem, we tried to gauge whether different CNN architectures trained using synthetic data were good enough to classify the weather captured in real time.

Transfer learning is a powerful machine learning technique, which allows re-usage of a model for different tasks. It has gained immense popularity for computer vision tasks where pretrained CNN architectures are used as the standard starting point given the vast resources in terms of computation and time required to develop CNNs from scratch.

The pipeline used for the work described in this paper is visually represented in Figure 7. The pipeline operates in a fashion where the weights of the entire pretrained network were frozen except the classification layers at the end. The softmax layer is added for multi weather classification. The softmax layer (4,1) is added because the number of classes is 4. The classifier layers of the pretrained networks were retrained on the proposed synthetic weather dataset. The test real-time images were passed through the retrained CNN models to extract predict the network’s accuracy.

The classifier layers were trained on the synthetic images and tested on the real-world dataset Berkeley DeepDrive [26]. After performing the experiment, the mean Average Precision (mAP) was calculated for each of the models.

### Pretrained Model

The pretrained models used for predicting weather are described in depth in the following subsections:**AlexNet**AlexNet [42] can easily be considered as a breakthrough network that has popularized deep learning approaches against traditional machine learning approaches. With eight layers, AlexNet won the famous object recognition challenge known as called the ImageNet Large Scale Visual Recognition Challenge (ILSVRC) in 2012. It is a variant of an artificial neural network, where the hidden layers comprise convolutional layers, pooling layers, fully connected layers, and normalization layers. A few of its standout features are the addition of nonlinearity, use of dropouts to overcome overfitting, and a reduction in network size due to overfitting.**VGGNET**VGGNET [43], a 19-layer network, was proposed as a step forward from AlexNet and was a runner up in the ILSVRC-2014 challenge. As an improvement, the large kernel size of the first and second convolutional layers of AlexNet net were replaced by multiple 3 × 3 size kernel filters. The small-size filters allow the network to have a large number of weight layers. Nonlinearity in decision making was incremented by adding 1 × 1 convolution layer.**GoogleLeNet**GoogleLeNet [44], a 22-layer network, was the winner of the ILSVRC-2014 challenge. It was proposed as a variant of an inception network to reduce the computational complexity of traditional CNNs.The inspection layer had variable receptive fields to capture sparse correlation patterns in the feature map.**Residual Network**Residual Network [45] was the winner of the ILSVRC-2015 challenge. It was proposed with the aim of overcoming the problem of a vanishing gradient in ultra-deep CNN by introducing residual blocks. Various versions of Residual Network (ResNet) were developed by varying the number of layers as 34, 50,101, 152, and 1202. The popular Residual Networks ResNet50 and ResNet101 were used in our experiment.

## 3. Results

In this section, we evaluate the various CNN models trained on our proposed synthetic dataset and compare their performance on the BDD dataset. The synthetic dataset contained images annotated with four weather classes. The number of epochs was set to 500. The learning rate of the stochastic gradient descent (SGD) optimizer for cross-entropy minimization was set to 0.0001. These parameters were deduced empirically by analyzing the training loss. As a regularization strategy during the training phase, two data augmentation techniques were used for all architectures. The first technique took random crops of training images, and the second technique applied rotation to the training images. All the algorithms were implemented using MATLAB, and the experiments were performed on a Tesla K80 with 12GB GPU memory and 916.77 GB storage.

Each experiment for calculating accuracy for the given pretrained model on the testing dataset was conducted 10 times. Then, the average accuracy for each model was calculated and denoted as mean Average Precision (mAP). The results tabulated in Table 2 show that the mAP for all the architectures varied between 60% and 74%. The accuracy variation over each epoch is shown in Figure 8 and Figure 9.

ResNet architectures achieved the lowest accuracy due to their complicated multi-branch designs, i.e., residual addition in ResNet, as the fine tuning of hyper-parameters and other customization becomes difficult. Given the constraint of hardware in self-driving cars, the inference is slowed down along with the reduction in memory utilization [46].

The most efficient weather classification accuracy on the testing dataset was achieved by the VGGNet architecture. These results indicate that the optimization achieved by the inclusion of smaller kernel filters at the initial convolutional layers had a positive effect on the overall task of weather classification. The universal effectiveness of the performance of VGGnet to extract deep features has also been affirmed previously by the state-of-the-art PFGFE-Net [13] that uses VGGNet as a backbone.

The training times from Table 2 reveal that they were directly proportional to the parameter due to the backpropagation process to retrain the weights of classification layers. However, with a closer look, one can conclude the training of the weather classification process for self driving cars was performed in the cloud, and it was a one time process. In the particular case of VGG, the training was time intensive, but it was a one time task. The testing time for determining weather from a single image on average using VGG was 15.67 fps, which is real-time efficient. Concluding the potential of this type of architecture on a classification task with a paucity of datasets draws attention to the possibility of more experimentation by training on a larger synthetic dataset with more diverse classes.

## 4. Conclusions

This paper highlighted the development of a custom driver simulator that was able to produce complex weather scenarios in immaculate detail. The manuscript also highlighted the possibility of using synthetic datasets to train a classifier in the context of weather classification and provided a synthetic dataset validated on the real-world Berkeley DeepDrive [26]. The proposed dataset was also hybrid in nature as synthetic images from different camera angles were taken. The weather classification accuracy was derived by testing classifiers on different real-time datasets, which allowed the persistent problem of bias in vision datasets to be tackled. The study proves that a persistent visual fidelity is important in generating realistic datasets for computer vision-based datasets. Furthermore, with advent of computer graphics it will be possible to achieve advanced photorealism in the generated datasets game engines such as Unity and Unreal are embedding new visualization techniques to further enable data scientists to generate accurate synthetic data for vision based tasks.

## Figures and Tables

**Figure 1 sensors-22-03193-f001:**
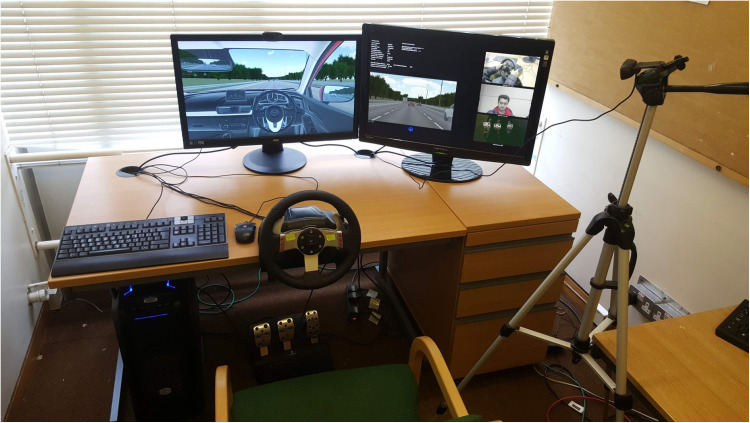
Simulator.

**Figure 2 sensors-22-03193-f002:**
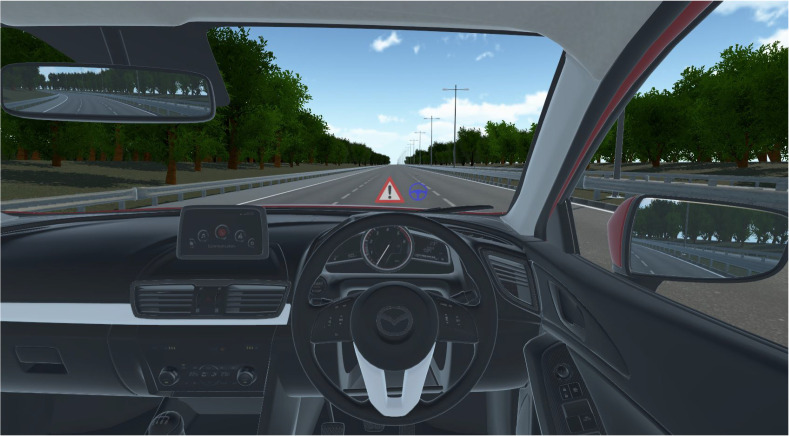
Virtual Interior.

**Figure 3 sensors-22-03193-f003:**
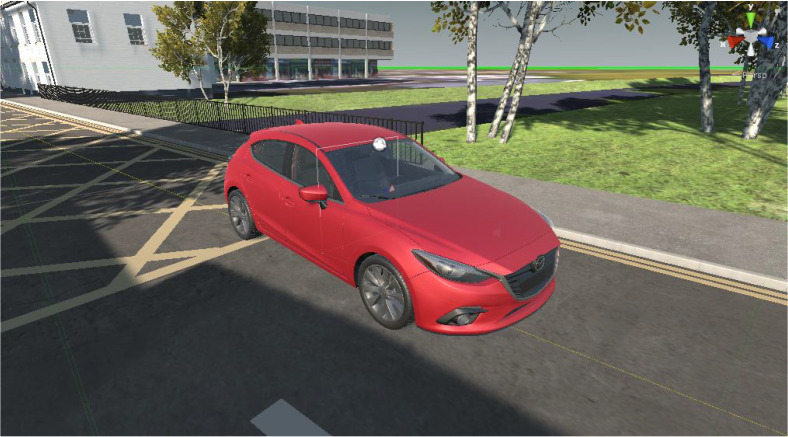
Virtual Car.

**Figure 4 sensors-22-03193-f004:**
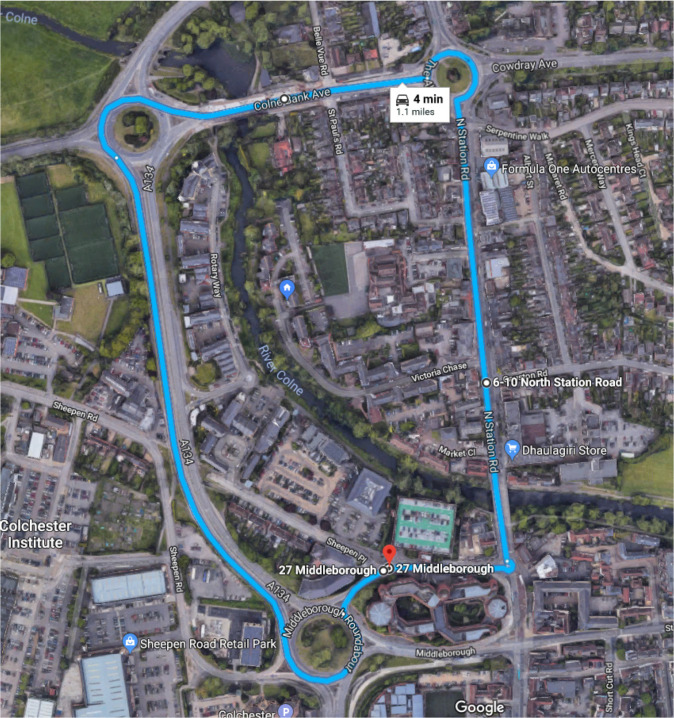
Proposed Environment.

**Figure 5 sensors-22-03193-f005:**
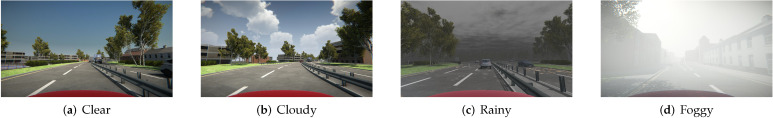
Synthetic Weather Dataset.

**Figure 6 sensors-22-03193-f006:**
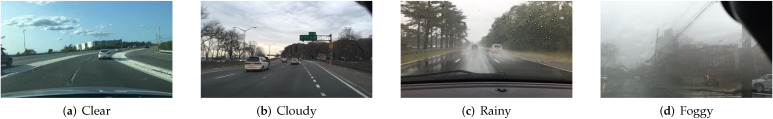
BDD (Berkeley Deep Dive) Dataset.

**Figure 7 sensors-22-03193-f007:**
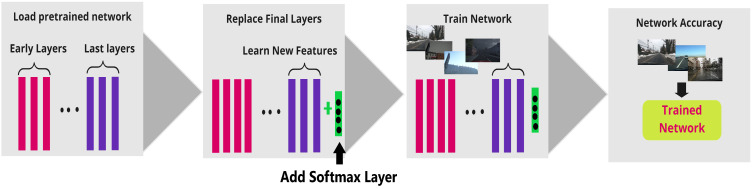
Pipeline: Step 1: Load the pretrained network, Step 2: Unfreeze the classification layers and add a softmax layer (4,1), Step 3: Train the weights of the classification layers with the synthetic dataset, Step 4: Test the network accuracy with a real time test dataset.

**Figure 8 sensors-22-03193-f008:**
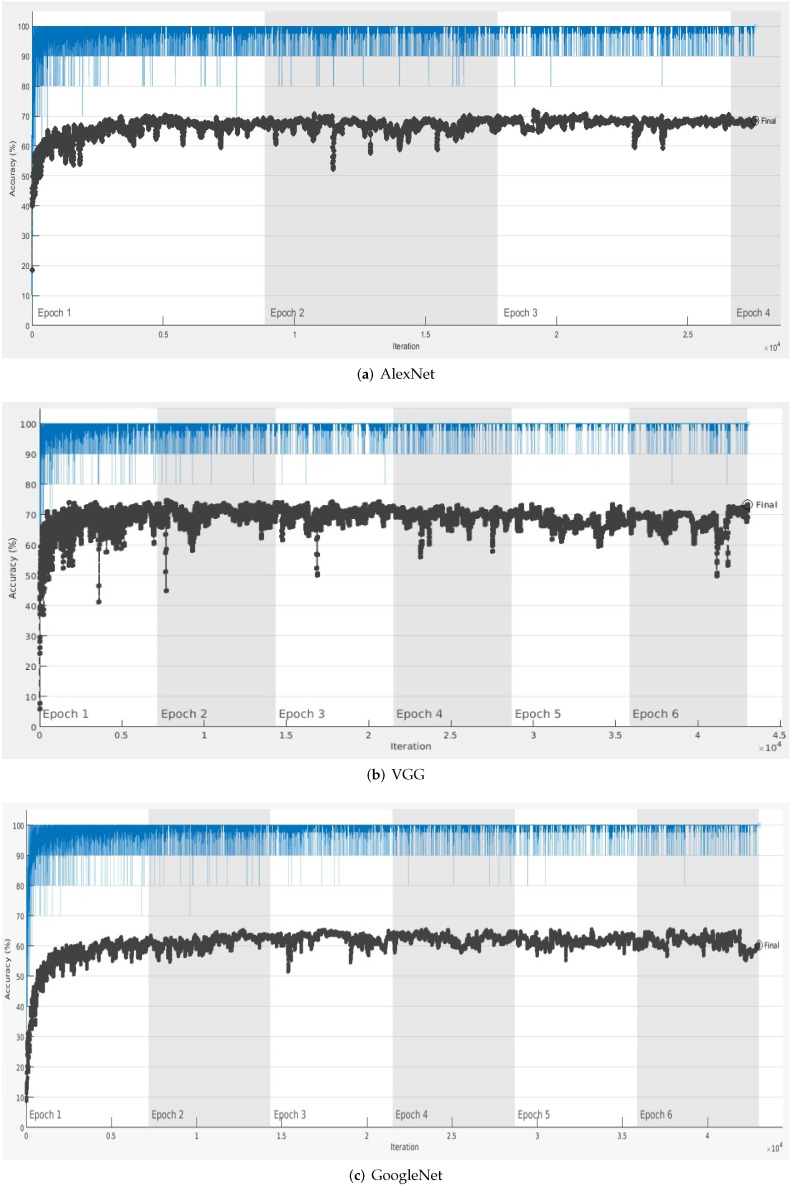
Accuracy variation over each epoch for (**a**) AlexNet, (**b**) VGG, and (**c**) GoogleLeNet models.

**Figure 9 sensors-22-03193-f009:**
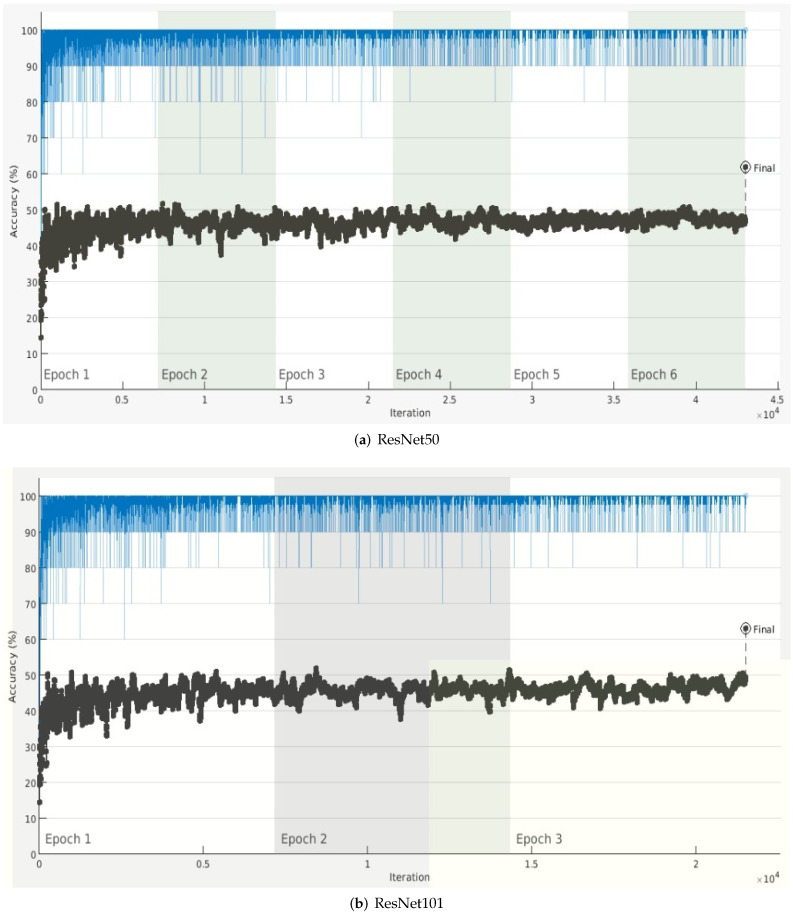
Accuracy variation over each epoch for Residual Networks (**a**) ResNet50 and (**b**) ResNet101.

**Table 1 sensors-22-03193-t001:** No. of Training images (Our dataset) and Testing images (BDD) per class distribution.

Class	Training	Testing
Clear	9613	1764
Cloudy	38,949	1677
Foggy	29,914	5
Rainy	29,857	396
Total	108,333	3842

**Table 2 sensors-22-03193-t002:** Results from CNN evaluations.

Architecture	mAP	Trainable Parameter	Time (min)
AlexNet	0.6856 ± 0.012	61M	986
VGGNET	0.7334 ± 0.023	138M	2930
GoogleLeNet	0.6034 ± 0.009	7M	618
ResNet50	0.6183 ± 0.025	26M	1020
ResNet101	0.63 ± 0.006	44M	1242

## Data Availability

The proposed weather dataset will be available freely in due course.

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
