# Peer review of "Weather Classification by Utilizing Synthetic Data"

_sensors, 2022, doi:10.3390/s22093193_

Round 1
Reviewer 1 Report
Authors conducted this research in the title of "Weather Classification by Utilizing Synthetic Data".
The paper’s subject could be interesting for readers of journal. Therefore, I recommend this paper for publication in this journal but before that, I have a few comments on the text that should be addressed before publication:
Comments:
1)Line 8, 9 and 10: There are too much words as Keywords in this section. Keywords could include less number of words than what it is now.
2) 1.1. Related Work: This title "1.1. Related Work" in page 2 of the article should be rewrited as "Related Works". Since authors probed several works in this section, the "s" is needed after the word "Work".
3) In line 4 of Abstract section authors used this word “We”. Words like “We”, “Me”, “Our” or “Us” are not common in article writing. Other words could be used by the authors. For example, this sentence "We presented a new model in this work" could be replaced by this "A new model is presented in this work".
4) In Abstract section authors did not mention the main goal of this research. In other words, there is no obvious words about the main question that this research is managed to answer it in this section. If it is added, it would be really helpful for readers to enter and understand main purpose of this research.
5) Conclusion: In the conclusion section authors should mention more words about their suggestions to future works related to this article. For example, how future works with similar titles can improve their accuracy of classification? Or which software could be used in future studies?
6) Which software has been used in this work to modelling data and export the results?. Also, which software has been used in this work to export the diagrams in this work?. For example, software like MATLAB could be used for modelling and software like SigmaPlot could be utilized to export charts and diagrams.
7) What is new in this research in comparison with other similar works? In other words, authors should write more about novelty of this research. They should talk more about their innovations in this article. For example, authors should answer this question "Is the used model in this work unprecedented and completely new?". This would be useful for readers to compare this article with other similar works conducted in recent years.
8) Figure 9 in page 7: The font of used numbers related to the axises of depicted charts in this figure should be bolder and more clear than what it is now. This could be really helpful for the readers of this article.
9) Conclusion: Conclusion section is too short in this article. This section should include more details. Below steps could be helpful for the authors to improve this section:
- Restate the thesis by making the same point with other words.
- Review your supporting ideas.
- For that, summarize all arguments by paraphrasing how you proved the thesis.
- Connect back to the essay hook and relate your closing statement to the opening one.
- Combine all the above to improved and expanded conclusion.
11) Since recently it has been proved that artificial intelligence (AI) and machine learning has a numerous applications in all of engineering fields, I highly recommend the authors to add some references in this manuscript in this regard. It would be useful for the readers of journal to get familiar with the application of AI in other engineering fields. I recommend the others to add all the following references, which are the newest references in this field
[1] Amidi, Y., Nazari, B., Sadri, S., & Yousefi, A. (2021). Parameter Estimation in Multiple Dynamic Synaptic Coupling Model Using Bayesian Point Process State-Space Modeling Framework. Neural Computation, 33(5), 1269-1299.
[2] Roshani, M., et al. 2020. Application of GMDH neural network technique to improve measuring precision of a simplified photon attenuation based two-phase flowmeter. Flow Measurement and Instrumentation, 75, p.101804.
[3] Yousefi, A., Amidi, Y., Nazari, B., & Eden, U. T. (2020). Assessing Goodness-of-Fit in Marked Point Process Models of Neural Population Coding via Time and Rate Rescaling. Neural Computation, 32(11), 2145-2186.
[4] Azizi, A., Tahmid, I. A., Waheed, A., Mangaokar, N., Pu, J., Javed, M., ... & Viswanath, B. (2021). T-Miner: A Generative Approach to Defend Against Trojan Attacks on DNN-based Text Classification. In 30th {USENIX} Security Symposium ({USENIX} Security 21).
Reviewer 2 Report
Line 16, the authors should and even must add the applications of CNN in various fields, (detection and classification) such as shipdenet-20, balance scene learning mechanism, hog-shipclsnet, squeeze-and-excitation laplacian pyramid network, quad-fpn, and PFGFE-Net.
Synthetic aperture radar is a tool for weather monitoring, please review them, for example, high-speed ship detection in sar images based on a grid convolutional neural network, depthwise separable convolution neural network for high-speed sar ship detection, ls-ssdd-v1.0, balance learning for ship detection from synthetic aperture radar remote sensing imagery, and HyperLi-Net.
Table 1, please explain why vggnet is the best. PFGFE-Net also used vgg (please review).
Figure 7, must add more network descriptions.
In figure 7, I cannot find the softmax, please add it from hog-shipclsnet, squeeze-and-excitation laplacian pyramid network, PFGFE-Net, and injection of traditional hand-crafted features into modern cnn-based models for sar ship classification: what, why, where, and how.
The authors should describe the dataset more clearly similar to sar ship detection dataset (ssdd): official release and comprehensive data analysis. Then, review this work.
The authors must consider the above comments and add them in suitable places in the main text, otherwise this paper cannot be accepted.
Round 2
Reviewer 1 Report
All comments have been addressed correctly
Reviewer 2 Report
This paper can be published in this state.